# GraphVAE: Towards Generation of Small Graphs Using Variational Autoencoders

## Abstract

Deep learning on graphs has become a popular research topic with many applications. However, past work has concentrated on learning graph embedding tasks only, which is in contrast with advances in generative models for images and text. Is it possible to transfer this progress to the domain of graphs? We propose to sidestep hurdles associated with linearization of such discrete structures by having a decoder output a probabilistic fully-connected graph of a predefined maximum size directly at once. Our method is formulated as a variational autoencoder. We evaluate on the challenging task of conditional molecule generation.

## 1 Introduction

Deep learning on graphs has very recently become a popular research topic, with useful applications across fields such as chemistry (Gilmer et al., 2017), medicine (Ktena et al.), or computer vision (Simonovsky & Komodakis, 2017). Past work has concentrated on learning graph embedding tasks so far, *i.e.* encoding an input graph into a vector representation. This is in stark contrast with fast-paced advances in generative models for images and text, which have seen massive rise in quality of generated samples. Hence, it is an intriguing question how one can transfer this progress to the domain of graphs, *i.e.* their decoding from a vector representation. Moreover, the desire for such a method has been mentioned in the past by Gómez-Bombarelli et al. (2016).

However, learning to generate graphs is a difficult problem for methods based on gradient optimization, as graphs are discrete structures. Incremental construction involves discrete decisions, which are not differentiable. Unlike sequence (text) generation, graphs can have arbitrary connectivity and there is no clear best way how to linearize their construction in a sequence of steps.

In this work, we propose to sidestep these hurdles by having the decoder output a probabilistic fully-connected graph of a predefined maximum size directly at once. In a probabilistic graph, the existence of nodes and edges, as well as their attributes, are modeled as independent random variables. The method is formulated in the framework of variational autoencoders (VAE) by Kingma & Welling (2013).

We demonstrate our method, coined GraphVAE, in cheminformatics on the task of molecule generation. Molecular datasets are a challenging but convenient testbed for our generative model, as they easily allow for both qualitative and quantitative tests of decoded samples. While our method is applicable for generating smaller graphs only and its performance leaves space for improvement, we believe our work is an important initial step towards powerful and efficient graph decoders.

## 2 Related work

**Graph Decoders.** Graph generation has been largely unexplored in deep learning. The closest work to ours is by Johnson (2017), who incrementally constructs a probabilistic (multi)graph as a world representation according to a sequence of input sentences to answer a query. While our model also outputs a probabilistic graph, we do not assume having a prescribed order of construction transformations available and we formulate the learning problem as an autoencoder.

Xu et al. (2017) learns to produce a scene graph from an input image. They construct a graph from a set of object proposals, provide initial embeddings to each node and edge, and use message passing

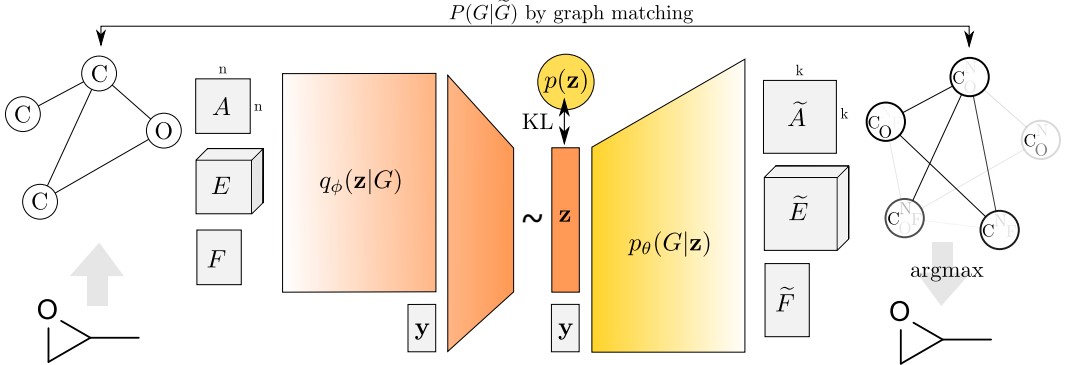

Figure 1: Illustration of the proposed variational graph autoencoder in its conditional form. Starting from a discrete attributed graph $G = (A, E, F)$ on $n$ nodes (*e.g.* a representation of propylene oxide), stochastic graph encoder $q_\phi(\mathbf{z}|G)$ embeds the graph into continuous representation $\mathbf{z}$. Given a point in the latent space, our novel graph decoder $p_\theta(G|\mathbf{z})$ outputs a probabilistic fully-connected graph $\widetilde{G} = (\widetilde{A}, \widetilde{E}, \widetilde{F})$ on predefined $k \geq n$ nodes, from which discrete samples may be drawn. The process can be conditioned on label $\mathbf{y}$ for controlled sampling at test time. Reconstruction ability of the autoencoder is facilitated by approximate graph matching for aligning $G$ with $\widetilde{G}$.

to obtain a consistent prediction. In contrast, our method is a generative model which produces a probabilistic graph from a single opaque vector, without specifying the number of nodes or the structure explicitly.

Related work pre-dating deep learning includes random graphs (Erdos & Rényi, 1960; Barabási & Albert, 1999), stochastic blockmodels (Snijders & Nowicki, 1997), or state transition matrix learning (Gong & Xiang, 2003).

**Discrete Data Decoders.** Text is the most common discrete representation. Generative models there are usually trained by teacher forcing (Williams & Zipser, 1989), which avoids the need to backpropagate through output discretization by feeding the ground truth instead of the past sample at each step. Recently, efforts have been made to overcome this problem. Notably, computing a differentiable approximation using Gumbel distribution (Kusner & Hernández-Lobato, 2016) or bypassing the problem by learning a stochastic policy in reinforcement learning (Yu et al., 2017). Our work also circumvents the non-differentiability problem, namely by formulating the loss on a probabilistic graph.

**Molecule Decoders.** Generative models may become promising for *de novo* design of molecules fulfilling certain criteria by being able to search for them over a continuous embedding space (Olive-crona et al., 2017). With that in mind, we propose a conditional version of our model. While molecules have an intuitive representation as graphs, the field has had to resort to textual representations with fixed syntax, *e.g.* so-called SMILES strings, to exploit recent progress made in text generation with RNNs (Olivecrona et al., 2017; Segler et al., 2017; Gómez-Bombarelli et al., 2016). As their syntax is brittle, many invalid strings tend to be generated, which has been recently addressed by Kusner et al. (2017) by incorporating grammar rules into decoding. While encouraging, their approach does not guarantee semantic (chemical) validity, similarly as our method.

## 3 METHOD

We approach the task of graph generation by devising a neural network able to translate vectors in a continuous code space to graphs. Our main idea is to output a probabilistic fully-connected graph and use a standard graph matching algorithm to align it to the ground truth. The proposed method is formulated in the framework of variational autoencoders (VAE) by Kingma & Welling (2013), although other forms of regularized autoencoders would be equally suitable (Makhzani et al., 2015;

Li et al., 2015a). We briefly recapitulate VAE below and continue with introducing our novel graph decoder together with an appropriate loss function.

## 3.1 VARIATIONAL AUTOENCODER

Let $G = (A, E, F)$ be a graph specified with its adjacency matrix $A$, edge attribute tensor $E$, and node attribute matrix $F$. We wish to learn an encoder and a decoder to map between the space of graphs $G$ and their continuous embedding $\mathbf{z} \in \mathbb{R}^c$, see Figure 1. In the probabilistic setting of a VAE, the encoder is defined by a variational posterior $q_\phi(\mathbf{z}|G)$ and the decoder by a generative distribution $p_\theta(G|\mathbf{z})$, where $\phi$ and $\theta$ are learned parameters. Furthermore, there is a prior distribution $p(\mathbf{z})$ imposed on the latent code representation as a regularization; we use a simplistic isotropic Gaussian prior $p(\mathbf{z}) = N(0, I)$. The whole model is trained by minimizing the upper bound on negative log-likelihood $-\log p_\theta(G)$ (Kingma & Welling, 2013):

$$\mathcal{L}(\phi, \theta; G) = \mathbb{E}_{q_\phi(\mathbf{z}|G)}[-\log p_\theta(G|\mathbf{z})] + \text{KL}[q_\phi(\mathbf{z}|G)||p(\mathbf{z})] \tag{1}$$

The first term of $\mathcal{L}$, the reconstruction loss, enforces high similarity of sampled generated graphs to the input graph $G$. The second term, KL-divergence, regularizes the code space to allow for sampling of $\mathbf{z}$ directly from $p(\mathbf{z})$ instead from $q_\phi(\mathbf{z}|G)$ later. The dimensionality of $\mathbf{z}$ is usually fairly small so that the autoencoder is encouraged to learn a high-level compression of the input instead of learning to simply copy any given input. While the regularization is independent on the input space, the reconstruction loss must be specifically designed for each input modality. In the following, we introduce our graph decoder together with an appropriate reconstruction loss.

## 3.2 PROBABILISTIC GRAPH DECODER

Graphs are discrete objects, ultimately. While this does not pose a challenge for encoding, demonstrated by the recent developments in graph convolution networks (Gilmer et al., 2017), graph generation has been an open problem so far. In a related task of text sequence generation, the currently dominant approach is character-wise or word-wise prediction (Bowman et al., 2016). However, graphs can have arbitrary connectivity and there is no clear way how to linearize their construction in a sequence of steps[1]. On the other hand, iterative construction of discrete structures during training without step-wise supervision involves discrete decisions, which are not differentiable and therefore problematic for back-propagation.

Fortunately, the task can become much simpler if we restrict the domain to the set of all graphs on maximum $k$ nodes, where $k$ is fairly small (in practice up to the order of tens). Under this assumption, handling dense graph representations is still computationally tractable. We propose to make the decoder output a probabilistic fully-connected graph $\widetilde{G} = (\widetilde{A}, \widetilde{E}, \widetilde{F})$ on $k$ nodes at once. This effectively sidesteps both problems mentioned above.

In probabilistic graphs, the existence of nodes and edges is modeled as Bernoulli variables, whereas node and edge attributes are multinomial variables. While not discussed in this work, continuous attributes could be easily modeled as Gaussian variables represented by their mean and variance. We assume all variables to be independent.

Each tensor of the representation of $\widetilde{G}$ has thus a probabilistic interpretation. Specifically, the predicted adjacency matrix $\widetilde{A} \in [0, 1]^{k \times k}$ contains both node probabilities $\widetilde{A}_{a,a}$ and edge probabilities $\widetilde{A}_{a,b}$ for nodes $a \neq b$. The edge attribute tensor $\widetilde{E} \in \mathbb{R}^{k \times k \times d_e}$ indicates class probabilities for edges and, similarly, the node attribute matrix $\widetilde{F} \in \mathbb{R}^{k \times d_n}$ contains class probabilities for nodes.

The decoder itself is deterministic. Its architecture is a simple multi-layer perceptron (MLP) with three outputs in its last layer. Sigmoid activation function is used to compute $\widetilde{A}$, whereas edge- and node-wise softmax is applied to obtain $\widetilde{E}$ and $\widetilde{F}$, respectively. At test time, we are often interested in a (discrete) point estimate of $\widetilde{G}$, which can be obtained by taking edge- and node-wise argmax in $\widetilde{A}$, $\widetilde{E}$, and $\widetilde{F}$. Note that this can result in a discrete graph on less than $k$ nodes.

---

[1]While algorithms for canonical graph orderings are available (McKay & Piperno, 2014), Vinyals et al. (2015) empirically found out that the linearization order matters when learning on sets.

### 3.3 RECONSTRUCTION LOSS

Given a particular of a discrete input graph $G$ on $n \leq k$ nodes and its probabilistic reconstruction $\widetilde{G}$ on $k$ nodes, evaluation of Equation 1 requires computation of likelihood $p_\theta(G|\mathbf{z}) = P(G|\widetilde{G})$.

Since no particular ordering of nodes is imposed in either $\widetilde{G}$ or $G$ and matrix representation of graphs is not invariant to permutations of nodes, comparison of two graphs is hard. However, approximate graph matching described further in Subsection 3.4 can obtain a binary assignment matrix $X \in \{0,1\}^{k \times n}$, where $X_{a,i} = 1$ only if node $a \in \widetilde{G}$ is assigned to $i \in G$ and $X_{a,i} = 0$ otherwise.

Knowledge of $X$ allows to map information between both graphs. Specifically, input adjacency matrix is mapped to the predicted graph as $A' = XAX^T$, whereas the predicted node attribute matrix and slices of edge attribute matrix are transferred to the input graph as $\widetilde{F}' = X^T \widetilde{F}$ and $\widetilde{E}'_{\cdot,\cdot,l} = X^T \widetilde{E}_{\cdot,\cdot,l} X$. The maximum likelihood estimates, *i.e.* cross-entropy, of respective variables are as follows:

$$
\begin{aligned}
\log p(A'|\mathbf{z}) = & 1/k \sum_a A'_{a,a} \log \widetilde{A}_{a,a} + (1 - A'_{a,a}) \log(1 - \widetilde{A}_{a,a}) + \\
& + 1/k(k-1) \sum_{a \neq b} A'_{a,b} \log \widetilde{A}_{a,b} + (1 - A'_{a,b}) \log(1 - \widetilde{A}_{a,b}) \\
\log p(F|\mathbf{z}) = & 1/n \sum_i \log F_{i,\cdot}^T \widetilde{F}'_{i,\cdot} \\
\log p(E|\mathbf{z}) = & 1/(||A||_1 - n) \sum_{i \neq j} \log E_{i,j,\cdot}^T \widetilde{E}'_{i,j,\cdot}
\end{aligned}
\tag{2}
$$

where we assumed that $F$ and $E$ are encoded in one-hot notation. The formulation considers existence of both matched and unmatched nodes and edges but attributes of only the matched ones. Furthermore, averaging over nodes and edges separately has shown beneficial in training as otherwise the edges dominate the likelihood. The overall reconstruction loss is a weighed sum of the previous terms:

$$
-\log p(G|\mathbf{z}) = -\lambda_A \log p(A'|\mathbf{z}) - \lambda_F \log p(F|\mathbf{z}) - \lambda_E \log p(E|\mathbf{z})
\tag{3}
$$

### 3.4 GRAPH MATCHING

The goal of (second-order) graph matching is to find correspondences $X \in \{0,1\}^{k \times n}$ between nodes of graphs $G$ and $\widetilde{G}$ based on the similarities of their node pairs $S : (i,j) \times (a,b) \to \mathbb{R}^+$ for $i, j \in G$ and $a, b \in \widetilde{G}$. It can be expressed as integer quadratic programming problem of similarity maximization over $X$ and is typically approximated by relaxation of $X$ into continuous domain: $X^* \in [0,1]^{k \times n}$ (Cho et al., 2014). For our use case, the similarity function is defined as follows:

$$
\begin{aligned}
S((i,j),(a,b)) = & (E_{i,j,\cdot}^T \widetilde{E}_{a,b,\cdot}) A_{i,j} \widetilde{A}_{a,b} \widetilde{A}_{a,a} \widetilde{A}_{b,b} [i \neq j \wedge a \neq b] + \\
& + (F_{i,\cdot}^T \widetilde{F}_{a,\cdot}) \widetilde{A}_{a,a} [i = j \wedge a = b]
\end{aligned}
\tag{4}
$$

The first term evaluates similarity between edge pairs and the second term between node pairs, $[\cdot]$ being the Iverson bracket. Note that the scores consider both feature compatibility ($\widetilde{F}$ and $\widetilde{E}$) and existential compatibility ($\widetilde{A}$), which has empirically led to more stable assignments during training. To summarize the motivation behind both Equations 3 and 4, our method aims to find the best graph matching and then further improve on it by gradient descent on the loss. Given the stochastic way of training deep network, we argue that solving the matching step only approximately is sufficient. This is conceptually similar to the approach for learning to output unordered sets by (Vinyals et al., 2015), where the closest ordering of the training data is searched for.

In practice, we are looking for a graph matching algorithm robust to noisy correspondences which can be easily implemented on GPU in batch mode. Max-pooling matching (MPM) by Cho et al.

(2014) is a simple but effective algorithm following the iterative scheme of power methods, see Appendix A for details. It can be used in batch mode if similarity tensors are zero-padded, *i.e.* $S((i, j), (a, b)) = 0$ for $n < i, j \leq k$, and the amount of iterations is fixed.

Max-pooling matching outputs continuous assignment matrix $X^*$. Unfortunately, attempts to directly use $X^*$ instead of $X$ in Equation 3 performed badly, as did experiments with direct maximization of $X^*$ or soft discretization with softmax or straight-through Gumbel softmax (Jang et al., 2016). We therefore discretize $X^*$ to $X$ using Hungarian algorithm to obtain a strict one-on-one mapping[2]. While this operation is non-differentiable, gradient can still flow to the decoder directly through the loss function and training convergence proceeds without problems. Note that this approach is often taken in works on object detection, *e.g.* (Stewart et al., 2016), where a set of detections need to be matched to a set of ground truth bounding boxes and treated as fixed before computing a differentiable loss.

### 3.5 FURTHER DETAILS

**Encoder.** A feed forward network with edge-conditioned graph convolutions (ECC) (Simonovsky & Komodakis, 2017) is used as encoder, although any other graph embedding method is applicable. As our edge attributes are categorical, a single linear layer for the filter generating network in ECC is sufficient. Due to smaller graph sizes no pooling is used in encoder except for global pooling, for which we employ soft attention pooling of Li et al. (2015b). As usual in VAE, we formulate encoder as probabilistic and enforce Gaussian distribution of $q_\phi(\mathbf{z}|G)$ by having the last encoder layer outputs $2c$ features interpreted as mean and variance, allowing to sample $\mathbf{z}_l \sim N(\mu_l(G), \sigma_l(G))$ for $l \in 1, .., c$ using the re-parameterization trick (Kingma & Welling, 2013).

**Disentangled Embedding.** In practice, rather than random drawing of graphs, one often desires more control over the properties of generated graphs. In such case, we follow Sohn et al. (2015) and condition both encoder and decoder on label vector $\mathbf{y}$ associated with each input graph $G$. Decoder $p_\theta(G|\mathbf{z}, \mathbf{y})$ is fed a concatenation of $\mathbf{z}$ and $\mathbf{y}$, while in encoder $q_\phi(\mathbf{z}|G, \mathbf{y})$, $\mathbf{y}$ is concatenated to every node's features just before the graph pooling layer. If the size of latent space $c$ is small, the decoder is encouraged to exploit information in the label.

**Limitations.** The proposed model is expected to be useful only for generating small graphs. This is due to growth of GPU memory requirements and number of parameters ($O(k^2)$) as well matching complexity ($O(k^4)$) with small decrease in quality for high values of $k$. In Section 4 we demonstrate results for up to $k = 38$. Nevertheless, for many applications even generation of small graphs is still very useful.

## 4 EVALUATION

We demonstrate our method for the task of molecule generation by evaluating on two large public datasets of organic molecules, QM9 and ZINC.

### 4.1 APPLICATION IN CHEMINFORMATICS

Quantitative evaluation of generative models of images and texts has been troublesome (Theis et al., 2015), as it very difficult to measure realness of generated samples in an automated and objective way. Thus, researchers frequently resort there to qualitative evaluation and embedding plots. However, qualitative evaluation of graphs can be very unintuitive for humans to judge unless the graphs are planar and fairly simple.

Fortunately, we found graph representation of molecules, as undirected graphs with atoms as nodes and bonds as edges, to be a convenient testbed for generative models. On one hand, generated graphs can be easily visualized in standardized structural diagrams. On the other hand, chemical validity of graphs, as well as many further properties a molecule can fulfill, can be checked using software

---

[2]Some predicted nodes are not assigned for $n < k$. Our current implementation performs this step on CPU although a GPU version has been published (Date & Nagi, 2016).

packages (`SanitizeMol` in RDKit) or simulations. This makes both qualitative and quantitative tests possible.

Chemical constraints on compatible types of bonds and atom valences make the space of valid graphs complicated and molecule generation challenging. In fact, a single addition or removal of edge or change in atom or bond type can make a molecule chemically invalid. Comparably, flipping a single pixel in MNIST-like number generation problem is of no issue.

To help the network in this application, we introduce three remedies. First, we make the decoder output symmetric $\widetilde{A}$ and $\widetilde{E}$ by predicting their (upper) triangular parts only, as undirected graphs are sufficient representation for molecules. Second, we use prior knowledge that molecules are connected and, at test time only, construct maximum spanning tree on the set of probable nodes $\{a : \widetilde{A}_{a,a} \geq 0.5\}$ in order to include its edges $(a, b)$ in the discrete pointwise estimate of the graph even if $\widetilde{A}_{a,b} < 0.5$ originally. Third, we do not generate Hydrogen explicitly and let it be added as "padding" during chemical validity check.

## 4.2 QM9 DATASET

QM9 dataset (Ramakrishnan et al., 2014) contains about 134k organic molecules of up to 9 heavy (non Hydrogen) atoms with 4 distinct atomic numbers and 4 bond types, we set $k = 9$, $d_e = 4$ and $d_n = 4$. We set aside 10k samples for testing and 10k for validation (model selection).

We compare our unconditional model to the character-based generator of Gómez-Bombarelli et al. (2016) (CVAE) and the grammar-based generator of Kusner et al. (2017) (GVAE). We used the code and architecture in Kusner et al. (2017) for both baselines, adapting the maximum input length to the smallest possible. In addition, we demonstrate a conditional generative model for an artificial task of generating molecules given a histogram of heavy atoms as 4-dimensional label $\mathbf{y}$, the success of which can be easily validated.

**Setup.** The encoder has two graph convolutional layers (32 and 64 channels) with identity connection, batchnorm, and ReLU; followed by soft attention pooling (Li et al., 2015b) with 128 channels and a fully-connected layer (FCL) to output $(\mu, \sigma)$. The decoder has 3 FCLs (128, 256, and 512 channels) with batchnorm and ReLU; followed by parallel triplet of FCLs to output graph tensors. We set $c = 40$, $\lambda_A = \lambda_F = \lambda_E = 1$, batch size 32, 75 MPM iterations and train for 25 epochs with Adam with learning rate 1e-3 and $\beta_1$=0.5.

**Embedding Visualization.** To visually judge the quality and smoothness of the learned embedding $\mathbf{z}$ of our model, we may traverse it in two ways: along a slice and along a line. For the former, we randomly choose two $c$-dimensional orthonormal vectors and sample $\mathbf{z}$ in regular grid pattern over the induced 2D plane. For the latter, we randomly choose two molecules $G^{(1)}, G^{(2)}$ of the same label from test set and interpolate between their embeddings $\mu(G^{(1)}), \mu(G^{(2)})$. This also evaluates the encoder, and therefore benefits from low reconstruction error.

We plot two planes in Figure 2, for a frequent label (left) and a less frequent label in QM9 (right). Both images show a varied and fairly smooth mix of molecules. The left image has many valid samples broadly distributed across the plane, as presumably the autoencoder had to fit a large portion of database into this space. The right exhibits stronger effect of regularization, as valid molecules tend to be only around center.

An example of several interpolations is shown in Figure 3. We can find both meaningful (1st, 2nd and 4th row) and less meaningful transitions, though many samples on the lines do not form chemically valid compounds.

**Decoder Quality Metrics.** The quality of a conditional decoder can be evaluated by the validity and variety of generated graphs. For a given label $\mathbf{y}^{(l)}$, we draw $n_s = 10^4$ samples $\mathbf{z}^{(l,s)} \sim p(\mathbf{z})$ and compute the discrete point estimate of their decodings $\hat{G}^{(l,s)} = \arg \max p_\theta(G|\mathbf{z}^{(l,s)}, \mathbf{y}^{(l)})$.

Let $V^{(l)}$ be the list of chemically valid molecules from $\hat{G}^{(l,s)}$ and $C^{(l)}$ be the list of chemically valid molecules with atom histograms equal to $\mathbf{y}^{(l)}$. We are interested in ratios $\text{Valid}^{(l)} = |V^{(l)}|/n_s$ and $\text{Accurate}^{(l)} = |C^{(l)}|/n_s$. Furthermore, let $\text{Unique}^{(l)} = |\text{set}(C^{(l)})|/|C^{(l)}|$ be the fraction of unique

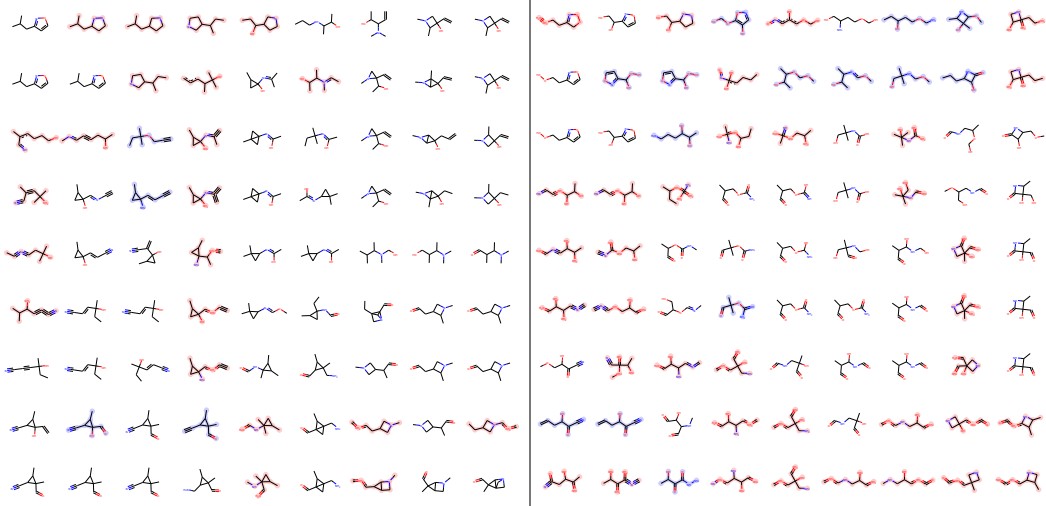

Figure 2: Decodings of latent space points sampled over a random 2D plane in **z**-space of $c = 40$ (within 5 units from center of coordinates). Left: Samples conditioned on 7x Carbon, 1x Nitrogen, 1x Oxygen (12% QM9). Right: Samples conditioned on 5x Carbon, 1x Nitrogen, 3x Oxygen (2.6% QM9). Color legend as in Figure 3.

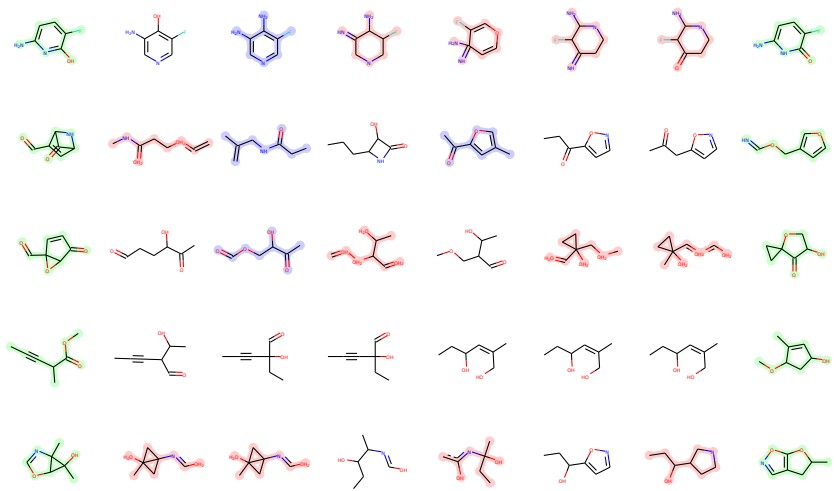

Figure 3: Linear interpolation between row-wise pairs of randomly chosen molecules in **z**-space of $c = 40$. Color legend: encoder inputs (green), chemically invalid graphs (red), valid graphs with wrong label (blue), valid and correct (white).

correct graphs and $\text{Novel}^{(l)} = 1 - |\text{set}(C^{(l)}) \cap \text{QM9}|/|\text{set}(C^{(l)})|$ the fraction of novel out-of-dataset graphs; we define $\text{Unique}^{(l)} = 0$ and $\text{Novel}^{(l)} = 0$ if $|C^{(l)}| = 0$. Finally, the introduced metrics are aggregated by frequencies of labels in QM9, *e.g.* $\text{Valid} = \sum_l \text{Valid}^{(l)}\text{freq}(\mathbf{y}^{(l)})$. Unconditional decoders are evaluated by assuming there is just a single label, therefore $\text{Valid} = \text{Accurate}$.

In Table 1, we can see that on average 50% of generated molecules are chemically valid and, in the case of conditional models, about 40% have the correct label which the decoder was conditioned on. Larger embedding sizes $c$ are less regularized, demonstrated by a higher number of $\text{Unique}$ samples and by lower accuracy of the conditional model, as the decoder is forced less to rely on actual labels. The ratio of $\text{Valid}$ samples shows less clear behavior, likely because the discrete performance is

|  |  | $\log p_\theta(G|\mathbf{z})$ | ELBO | Valid | Accurate | Unique | Novel |
|---|---|---|---|---|---|---|---|
| Cond. | Ours $c = 20$ | -0.578 | -0.722 | 0.565 | 0.467 | 0.314 | 0.598 |
|  | Ours $c = 40$ | -0.504 | -0.617 | 0.511 | 0.416 | 0.484 | 0.635 |
|  | Ours $c = 60$ | -0.492 | -0.585 | 0.520 | 0.406 | 0.583 | 0.613 |
|  | Ours $c = 80$ | -0.475 | -0.557 | 0.458 | 0.353 | 0.666 | 0.661 |
| Unconditional | Ours $c = 20$ | -0.660 | -0.916 | 0.485 | 0.485 | 0.457 | 0.575 |
|  | Ours $c = 40$ | -0.537 | -0.744 | 0.542 | 0.542 | 0.618 | 0.617 |
|  | Ours $c = 60$ | -0.486 | -0.656 | 0.517 | 0.517 | 0.695 | 0.570 |
|  | Ours $c = 80$ | -0.482 | -0.628 | 0.557 | 0.557 | 0.760 | 0.616 |
|  | NoGM $c = 80$ | -2.388 | -2.553 | 0.810 | 0.810 | 0.241 | 0.610 |
|  | CVAE $c = 60$ | – | – | 0.103 | 0.103 | 0.675 | 0.900 |
|  | GVAE $c = 20$ | – | – | 0.602 | 0.602 | 0.093 | 0.809 |

Table 1: Performance on conditional and unconditional QM9 models evaluated by mean test-time reconstruction log-likelihood ($\log p_\theta(G|\mathbf{z})$), mean test-time evidence lower bound (ELBO), and decoding quality metrics (Section 4.2). Baselines CVAE (Gómez-Bombarelli et al., 2016) and GVAE(Kusner et al., 2017) are listed only for the embedding size with the highest Valid.

not directly optimized for. For all models, it is remarkable that about 60% of generated molecules are out of the dataset, *i.e.* the network has never seen them during training. In Appendix B we additionally trade uniqueness for validity.

Looking at the baselines, CVAE can output only very few valid samples as expected, while GVAE generates the highest number of valid samples (60%) but of very low variance (less than 10%). Additionally, we investigate the importance of graph matching by using identity assignment $X$ instead and thus learning to reproduce particular node permutations in the training set, which correspond to the canonical ordering of SMILES strings from rdkit. This ablated model (denoted as NoGM in Table 1) produces many valid samples of lower variety and, surprisingly, outperforms GVAE in this regard. In comparison, our model can achieve good performance in both metrics at the same time.

**Likelihood.** Besides the application-specific metric introduced above, we also report evidence lower bound (ELBO) commonly used in VAE literature, which corresponds to $-\mathcal{L}(\phi, \theta; G)$ in our notation. In Table 1, we state mean bounds over train and test set, using a single $\mathbf{z}$ sample per graph. We observe both reconstruction loss and KL-divergence decrease due to larger $c$ providing more freedom. However, there seems to be no strong correlation between ELBO and Valid, which makes model selection somewhat difficult.

### 4.3 ZINC DATASET

ZINC dataset (Irwin et al., 2012) contains about 250k drug-like organic molecules of up to 38 heavy atoms with 9 distinct atomic numbers and 4 bond types, we set $k = 38$, $d_e = 4$ and $d_n = 9$ and use the same split strategy as with QM9. We investigate the degree of scalability of an unconditional generative model.

**Setup.** The setup is equivalent as for QM9 but with a wider encoder (64, 128, 256 channels).

**Decoder Quality Metrics.** Our best model with $c = 40$ has archived Valid $= 0.135$, which is clearly worse than for QM9. For comparison, CVAE failed to generated any valid sample, while GVAE achieved Valid $= 0.357$ (models provided by Kusner et al. (2017), $c = 56$).

We attribute such a low performance to a generally much higher chance of producing a chemically-relevant inconsistency (number of possible edges growing quadratically). To confirm the relationship between performance and graph size $k$, we kept only graphs not larger than $k = 20$ nodes, corresponding to 21% of ZINC, and obtained Valid $= 0.341$ (and Valid $= 0.185$ for $k = 30$ nodes, 92% of ZINC). To verify that the problem is likely not caused by our proposed graph matching loss, we synthetically evaluate it in the following.

| Noise | $k = 15$ | $k = 20$ | $k = 25$ | $k = 30$ | $k = 35$ | $k = 40$ |
|---|---|---|---|---|---|---|
| $\epsilon_{A,E,F} = 0$ | 99.55 | 99.52 | 99.45 | 99.4 | 99.47 | 99.46 |
| $\epsilon_A = 0.4$ | 90.95 | 89.55 | 86.64 | 87.25 | 87.07 | 86.78 |
| $\epsilon_A = 0.8$ | 82.14 | 81.01 | 79.62 | 79.67 | 79.07 | 78.69 |
| $\epsilon_E = 0.4$ | 97.11 | 96.42 | 95.65 | 95.90 | 95.69 | 95.69 |
| $\epsilon_E = 0.8$ | 92.03 | 90.76 | 89.76 | 89.70 | 88.34 | 89.40 |
| $\epsilon_F = 0.4$ | 98.32 | 98.23 | 97.64 | 98.28 | 98.24 | 97.90 |
| $\epsilon_F = 0.8$ | 97.26 | 97.00 | 96.60 | 96.91 | 96.56 | 97.17 |

Table 2: Mean accuracy of matching ZINC graphs to their noisy counterparts in a synthetic benchmark as a function of maximum graph size $k$.

**Matching Robustness.** Robust behavior of graph matching using our similarity function $S$ is important for good performance of GraphVAE. Here we study graph matching in isolation to investigate its scalability. To that end, we add Gaussian noise $N(0, \epsilon_A), N(0, \epsilon_E), N(0, \epsilon_F)$ to each tensor of input graph $G$, truncating and renormalizing to keep their probabilistic interpretation, to create its noisy version $G_N$. We are interested in the quality of matching between self, $P[G, G]$, using noisy assignment matrix $X$ between $G$ and $G_N$. The advantage to naive checking $X$ for identity is the invariance to permutation of equivalent nodes.

In Table 2 we vary $k$ and $\epsilon$ for each tensor separately and report mean accuracies (computed in the same fashion as losses in Equation 3) over 100 random samples from ZINC with size up to $k$ nodes. While we observe an expected fall of accuracy with stronger noise, the behavior is fairly robust with respect to increasing $k$ at a fixed noise level, the most sensitive being the adjacency matrix. Note that accuracies are not comparable across tables due to different dimensionalities of random variables. We may conclude that the quality of the matching process is not a major hurdle to scalability.

## 5  CONCLUSION

In this work we addressed the problem of generating graphs from a continuous embedding in the context of variational autoencoders. We evaluated our method on two molecular datasets of different maximum graph size. While we achieved to learn embedding of reasonable quality on small molecules, our decoder had a hard time capturing complex chemical interactions for larger molecules. Nevertheless, we believe our method is an important initial step towards more powerful decoders and will spark interesting in the community.

There are many avenues to follow for future work. Besides the obvious desire to improve the current method (for example, by incorporating a more powerful prior distribution or adding a recurrent mechanism for correcting mistakes), we would like to extend it beyond a proof of concept by applying it to real problems in chemistry, such as optimization of certain properties or predicting chemical reactions. An advantage of a graph-based decoder compared to SMILES-based decoder is the possibility to predict detailed attributes of atoms and bonds in addition to the base structure, which might be useful in these tasks. Our autoencoder can also be used to pre-train graph encoders for fine-tuning on small datasets (Goh et al., 2017).

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

## APPENDIX

## A  MAX-POOLING MATCHING

In this section we briefly review max-pooling matching algorithm of Cho et al. (2014). In its relaxed form, a continuous correspondence matrix $X^* \in [0,1]^{k \times n}$ between nodes of graphs $G$ and $\widetilde{G}$ is determined based on similarities of node pairs $i, j \in G$ and $a, b \in \widetilde{G}$ represented as matrix elements $S_{ia;jb} \in \mathbb{R}^+$.

Let $\mathbf{x}^*$ denote the column-wise replica of $X^*$. The relaxed graph matching problem is expressed as quadratic programming task $\mathbf{x}^* = \arg\max_{\mathbf{x}} \mathbf{x}^T S \mathbf{x}$ such that $\sum_{i=1}^n \mathbf{x}_{ia} \le 1$, $\sum_{a=1}^k \mathbf{x}_{ia} \le 1$, and $\mathbf{x} \in [0,1]^{kn}$. The optimization strategy of choice is derived to be equivalent to the power method with iterative update rule $\mathbf{x}^{(t+1)} = S\mathbf{x}^{(t)}/||S\mathbf{x}^{(t)}||_2$. The starting correspondences $\mathbf{x}^{(0)}$ are initialized as uniform and the rule is iterated until convergence; in our use case we run for a fixed amount of iterations.

In the context of graph matching, the matrix-vector product $S\mathbf{x}$ can be interpreted as sum-pooling over match candidates: $\mathbf{x}_{ia} \leftarrow \mathbf{x}_{ia}S_{ia;ia} + \sum_{j \in N_i} \sum_{b \in N_a} \mathbf{x}_{jb}S_{ia;jb}$, where $N_i$ and $N_a$ denote the set of neighbors of node $i$ and $a$. The authors argue that this formulation is strongly influenced by uninformative or irrelevant elements and propose a more robust max-pooling version, which considers only the best pairwise similarity from each neighbor: $\mathbf{x}_{ia} \leftarrow \mathbf{x}_{ia}S_{ia;ia} + \sum_{j \in N_i} \max_{b \in N_a} \mathbf{x}_{jb}S_{ia;jb}$.

|  |  | $\log p_\theta(G|\mathbf{z})$ | ELBO | Valid | Accurate | Unique | Novel |
|---|---|---|---|---|---|---|---|
| Cond. | Ours/imp $c = 20$ | -0.784 | -0.919 | *0.572* | *0.482* | 0.238 | *0.718* |
|  | Ours/imp $c = 40$ | -0.671 | -0.776 | *0.611* | *0.518* | 0.307 | *0.665* |
|  | Ours/imp $c = 60$ | -0.618 | -0.714 | *0.566* | *0.448* | 0.416 | *0.710* |
|  | Ours/imp $c = 80$ | -0.627 | -0.713 | *0.583* | *0.451* | 0.475 | *0.681* |
| Uncond. | Ours/imp $c = 20$ | -0.857 | -1.091 | *0.533* | *0.533* | 0.228 | *0.610* |
|  | Ours/imp $c = 40$ | -0.737 | -0.932 | *0.562* | *0.562* | 0.420 | *0.758* |
|  | Ours/imp $c = 60$ | -0.634 | -0.797 | *0.587* | *0.587* | 0.459 | *0.730* |
|  | Ours/imp $c = 80$ | -0.642 | -0.777 | *0.571* | *0.571* | 0.520 | *0.719* |

Table 3: Performance on conditional and unconditional QM9 models with implicit node probabilities. Improvement with respect to Table 1 is emphasized in italics.

## B  IMPLICIT NODE PROBABILITIES

Our decoder assumes independence of node and edge probabilities, which allows for isolated nodes or edges. Making further use of the fact that molecules are connected graphs, we investigate the effect of making node probabilities a function of edge probabilities in this section. Specifically, we define the probability for node $a$ as that of its most probable edge: $\widetilde{A}_{a,a} = \max_b \widetilde{A}_{a,b}$.

The evaluation on QM9 in Table 3 shows a clear improvement in Valid, Accurate, and Novel metrics in both the conditional and unconditional setting. However, this is paid for by lower variability and higher reconstruction loss. This indicates that while the new constraint is useful, the model cannot fully cope with it. Moreover, we have seen no improvement on ZINC dataset.

## C  UNREGULARIZED AUTOENCODER

The regularization in VAE works against achieving perfect reconstruction of training data, especially for small embedding sizes. To understand the reconstruction ability of our architecture, we train it as unregularized in this section, *i.e.* with a deterministic encoder and without KL-divergence term in Equation 1.

Unconditional models for QM9 achieve mean test log-likelihood $\log p_\theta(G|\mathbf{z})$ of roughly $-0.37$ (about $-0.50$ for the implicit model in Appendix B) for all $c \in \{20, 40, 60, 80\}$. While these log-likelihoods are significantly higher than in Tables 1 and 3, our architecture can not achieve perfect reconstruction of inputs. We were successful to increase training log-likelihood to zero only on fixed small training sets of hundreds of examples, where the network could overfit. This indicates that the network has problems finding generally valid rules for assembly of output tensors.

