# OpenReview forum: "GraphVAE: Towards Generation of Small Graphs Using Variational Autoencoders"
_ICLR.cc/2018/Conference — Reject_

### Official Review · AnonReviewer1 · 2017-11-25
**An interesting Graph Generator**

**Rating:** 7
**Confidence:** 2

**Review:**

This work proposed an interesting graph generator using a variational autoencoder. The work should be interesting to researchers in the various areas. However, the work can only work on small graphs. The search space of small graph generation is usually very small, is there any other traditional methods can work on this problem? Moreover, the notations are a little confusing.

---

> ### Author Response · Authors · 2017-12-23
> **Rebuttal**
>
> Thank you for your review. Regarding traditional methods besides stochastic blockmodels [Snijders & Nowicki, 1997], we should have also mentioned the research on random graph models, such as [Erdös & Rényi, 1960] or [Barabási & Albert, 1999]. These models make fairly strong assumptions and cannot be used to model e.g. chemical compounds, though. You can consult e.g. "A Survey of Statistical Network Models" [Goldenberg et al, 2009] for a detailed review.
>
> Regarding the little confusing notation, we have updated the paper in several places today; could you please provide more details so that we can further improve the manuscript?

---

### Official Review · AnonReviewer2 · 2017-12-01
**interesting step in generative ANN architectures  for graphs**

**Rating:** 7
**Confidence:** 4

**Review:**

The authors propose a variational auto encoder architecture to generate graphs.

Pros:
- the formulation of the problem as the modeling of a probabilistic graph is of interest
- some of the main issues with graph generation are acknowledged (e.g. the problem of invariance to node permutation) and a solution is proposed (the binary assignment  matrix)
- notions for measuring the quality of the output graphs are of interest: here the authors propose some ways to use domain knowledge to check simple properties of molecular graphs

Cons:
- the work is quite preliminary
- many crucial elements  in graph generation are not dealt with:
 a) the adjacency matrix and the label tensors are not independent of each other, the notion of a graph is in itself a way to represent the 'relational links' between the various components
 b) the boundaries between a feasible and an infeasible graph are sharp: one edge or one label can be sufficient for acting the transition independently of the graph size, this makes it a difficult task for a continuous model. The authors acknowledge this but do not offer ways to tackle the issue
 c) conditioning on the label histogram should make the problem easy: one is giving away the number of nodes and the label identities after all; however even in this setup the approach fails more often than not
 d) the graph matching procedure proposed is a rough patch for a much deeper problem
- the evaluation should include a measure of the capacity of the architecture to :
 a) reconstruct perfectly the input
 b) denoise perturbations over node labels and additional/missing  edges

---

> ### Author Response · Authors · 2017-12-23
> **Rebuttal**
>
> Thank you for your review. We address your critique in the following.
>
> # a) adjacency matrix and the label tensors are not independent of each other
>
> Our decoder uses a single stream of feature channels until its last layer, which should make the three predicted tensors rather dependent. In fact, we tried to go a step further and derive the adjacency matrix from feature tensors by introducing a virtual "not-present" edge and node class. However, this did not improve performance, likely due to a the fact that this required whole feature tensors to be correct, whereas our presented loss ignores unmatched parts of these tensors.
>
> # c) conditioning on the label histogram should make the problem easy: one is giving away the number of nodes and the label identities after all; however even in this setup the approach fails more often than not
>
> Thank you for making this hypothesis. We performed an additional experiment by training in unconditioned setting on QM9 (see updated Table 1). Indeed, conditional training is able to reach a lower loss, though this difference diminishes with increasing size of the embedding (likely due to the autoencoder having more freedom to capture such statistics by itself). The number of valid samples fluctuates over configurations and is roughly the same for both conditional and unconditional setting.
>
> We managed to improve our results on QM9 (so that can it succeed slightly more often than not), compared them to previous work, and found that the ratio of valid samples can be similar to a grammar-based decoder [Kusner et al, 2017] on QM9 while offering much higher variance; see Tables 1 and 3 in the updated paper. Unlike Kusner et al, we could achieve this without manual specification of a grammar or other rules, besides the help from maximum spanning tree.
>
> # A measure of the capacity of the architecture to reconstruct perfectly the input
>
> This is a very good point. To this end, we removed the regularization and trained our architecture as a standard autoencoder, where the only goal is to aim for perfect reconstruction. Unfortunately, it turned out the architecture is not powerful enough to perfectly reconstruct the input, unless the set of possible inputs is rather small (e.g. for a fixed set of 1000 training examples). We added this information to Appendix C. In this light, we did not pursue the scenario of denoising autoencoder, which you also suggested.

---

### Official Review · AnonReviewer3 · 2017-12-04
**Interesting topic, but paper isn't ready yet**

**Rating:** 5
**Confidence:** 3

**Review:**

This paper studies the problem of learning to generate graphs using deep learning methods. The main challenges of generating graphs as opposed to text or images are said to be the following:
(a) Graphs are discrete structures, and incrementally constructing them would lead to non-differentiability (I don't agree with this; see below)
(b) It's not clear how to linearize the construction of graphs due to their symmetries. Based on this motivation, the paper decides to generate a graph in "one shot", directly  outputting node and edge existence probabilities, and node attribute vectors.

A graph is represented by a soft adjacency matrix A (entries are probability of existence of an edge), an edge attribute tensor E (entries are probability of each edge being one of d_e discrete types), and a node attribute matrix F, which has a node vector for each  potential node. A cross entropy loss is developed to measure the loss between generated A, E, and F and corresponding targets.

The main issue with training models in this formulation is the alignment of the generated graph to the ground truth graph. To handle this, the paper proposes to use a simple graph  matching algorithm (Max Pooling Matching) to align nodes and edges. A downside to the algorithm is that it has complexity O(k^4) for graphs with k nodes, but the authors argue that this is not a problem when generating small graphs. Once the best correspondence is found, it is treated as constant and gradients are propagated appropriately.

Experimentally, generative models of chemical graphs are trained on two datasets. Qualitative results and ELBO values are reported as the dimensionality of the embeddings is varied. No baseline results are presented. A further small set of experiments evaluates the quality of the matching algorithm on a synthetic setup.

Strengths:
- Generating graphs is an interesting problem, and the proposed approach seems like an easy-to-implement, mostly reasonable way of approaching the problem.

- The exposition is clear (although a bit more detail on MPM matching would be appreciated)

However, there are some significant weaknesses. First, the motivation for one-shot graph construction is not very strong:

- I don't understand why the non-differentiability argued in (a) above is an issue. If training uses a maximum likelihood objective, then we should be able to decompose the generation of a graph into a sequence of decisions and maximize the sum of the logprobs of the conditionals. People do this all the time with sequence data and non-differentiability is not an issue.

- I also don't agree that the one shot graph construction sidesteps the issue of how to linearize the construction of a graph. Even after doing so, the authors need to solve a matching problem to resolve the alignment issue. I see this as equivalent to choosing an order in which to linearize the order of nodes and edges in the graph.

Second, the experiments are quite weak. No baselines are presented to back up the claims motivating the formulation. I don't know how to interpret whether the results are good or bad. I would have at least liked to see a comparison to a method that generated SMILES format in an autoregressive manner (similar to previous work on chemical graph generation), and would ideally have liked to see an attempt at solving the alignment problem within an autoregressive formulation (e.g., by greedily constructing the alignment as the graph was generated). If one is willing to spend O(k^4) computation to solve the alignment problem, then there seem like many possibilities that could be easily applied to the autoregressive formulation. The authors might also be interested in a concurrent ICLR submission that approaches the problem from an autoregressive angle (https://openreview.net/pdf?id=Hy1d-ebAb).

Finally, I would have expected to see a discussion and comparison to "Learning Graphical State Transitions" (Johnson, 2017). Please also don't make statements like "To the best of our knowledge, we are the first to address graph generation using deep learning." This is very clearly not true. Even disregarding Johnson (2017), which the authors claim to be unaware of, I would consider approaches that generate SMILES format (like Gomez-Bombarelli et al) to be doing graph generation using deep learning.

Overall, the paper is about an interesting subject, but in my opinion the execution isn't strong enough to warrant publication at this point.

---

> ### Author Response · Authors · 2017-12-23
> **Rebuttal**
>
> Thank you for your review. We address your critique in the following.
>
> # Ordering vs Alignment
>
> The choice between linearization and matching is certainly an interesting topic, these are indeed two sides of the same coin. The graph canonization problem (i.e. consistent node ordering) is at least as computationally hard as the graph isomorphism problem (i.e. matching), which is NP-hard for general graphs. Fortunately, there are practical algorithms available for both problems, such as Nauty [McKay & Piperno, 2014] for canonization and max-pooling matching [Cho et al, 2014] for approximate isomorphism, used in our paper. Thus, both ways are feasible for small graphs, though not for free.
>
> We decided for one-shot construction with matching to allow the decoder to find its own orderings, motivated by the empirical result of [Vinyals et al, ICLR'16] that the linearization order matters when learning on sets. It is a priori unclear that enforcing a specific canonical ordering of vertices with a strategy for incremental construction (e.g. adding vertices one by one and connecting them to existing nodes) would lead to the best results. In this sense, we indeed sidestep the issue of how to linearize the construction by postponing the correspondence problem to the loss for the final result. We do not avoid the computational penalty of alignment. Note that our matching approach can be seen as inexact search of output permutation in Equation 9 in [Vinyals et al, ICLR'16].
>
> One could also consider incremental (likely autoregressive, as you suggested) construction with matching. However, [Johnson, ICLR'17] noted in his construction of probabilistic graphs that a loss function for only the final result was insufficient and deep supervision over individual construction steps was necessary for good performance. Your idea of "greedily constructing the alignment as the graph was generated" certainly sounds quite promising in this context, thank you for it. It might nicely combine the idea of the concurrent submission (https://openreview.net/pdf?id=Hy1d-ebAb) and our paper. Though we would consider it as a direction for future work at this momement, as it would lead to extensive modification of our current submission.
>
>
> # Non-differentiability
>
> We agree that non-differentiability is not a major obstacle if the generation of a graph is linearized, i.e. decomposed into a sequence of decisions in the ground truth. This may be given by the nature of some tasks, such as those addressed by [Johnson, ICLR'17], where graphs are built according to a sequence of statements. In general, however, the choice of such a decomposition is not clear, as we argue above. In this regard, it is interesting to learn from the mentioned concurrent submission (https://openreview.net/pdf?id=Hy1d-ebAb) that random orderings seem to work well. Nevertheless, even ML training with teacher forcing is not the perfect solution due to exposure bias (possibly poor prediction performance if the RNN's conditioning context diverges from sequences seen during training, i.e. the inability to fix its mistakes) [Bengio et al, 2015].
>
> # Baselines
>
> We agree that the omission of baselines was clearly a weak point. In the updated paper, we compare with character-based decoder of [Gomez-Bombarelli et al, 2016] and grammar-based decoder of [Kusner et al, 2017]. We found that the ratio of valid samples can be similar to a grammar-based decoder [Kusner et al, 2017] on QM9 while offering much higher variance; see Tables 1 and 3 in the updated paper. Unlike Kusner et al, we could achieve this without manual specification of a grammar or other rules, besides the help from maximum spanning tree.
>
> # Other points
>
> Thank you for the reference to [Johnson, ICLR'17], we have updated the paper in this regard and toned down our statement on being the first, in this light. We also included a short appendix on MPM matching.

---

> > ### Comment · AnonReviewer3 · 2018-01-17
> > **Updated score based on additional experiments and discussion**
> >
> > Thanks for the response, adding baselines, and a better treatment of related work. I've raised my score by a point.

---

> > > ### Author Response · Authors · 2018-01-17
> > > **Re: Updated score**
> > >
> > > Thank you!

---

### Public Comment · (anonymous) · 2017-12-02
**Reference to previous work on matching via the Hungarian algorithm**

I would like to point the authors to a relevant paper that similarly solves a one-to-one matching problem on unordered sets via the Hungarian algorithm within an end-to-end model:

R. Stewart, M. Andriluka, A.Y. Ng, End-to-End People Detection in Crowded Scenes, CVPR 2016

I think it would be fair to cite and discuss their work.

---

> ### Author Response · Authors · 2017-12-23
> **Re: previous work**
>
> Thank you for making us aware of the connection to object detection literature. We have added a reference to Stewart et al. in the updated version of our paper. Indeed, we share the same problem of matching unordered network outputs to ground truth, although the matching freedom is additionally constrained by edges in our case and we need to consider this by first running approximate graph matching to get reasonable similarities for Hungarian algorithm. As in our submission, Stewart et al. assumes that the matching is fixed for a given iteration and the gradient does not flow through the actual computation of matching (it is therefore not a perfect end-to-end model). Using a fixed matching in loss functions appears also in earlier (deep learning based) object detection papers in fact, e.g. Scalable High Quality Object Detection by Szegedy et al., 2014 (https://arxiv.org/abs/1412.1441).

---

### Author Response · Authors · 2017-12-23
**Summary of updated version**

- We added comparison to two baselines ([Kusner et al, 2017] and [Gomez-Bombarelli et al, 2016]) on QM9 and ZINC, results of unconditioned models on QM9 (Table 1), and results with unregularized training (Appendix C).
- We introduce a model variant with a higher percentage of valid samples by making node probabilities a function of edge probabilities (Appendix B).
- We added a brief summary of max-pooling matching (Appendix A).
- We made multiple minor edits over the paper to enhance clarity, mention further observations, and refer to more related work such as [Johnson, 2017], [Vinyals et al, 2016], and [Stewart et al, 2016].

---

### Public Comment · (anonymous) · 2018-01-12
**Request for clarification on pooling layer and/or source code**

I've tried implementing the model described in the paper, but I cannot understand why a global pooling layer would need channels. The cited paper (Li et al., 2015b) doesn't really seem to be related to pooling, and doesn't even mention it at all.

Could you elaborate a bit on the matter, or point me to an implementation of such a pooling layer?

Thanks

---

> ### Author Response · Authors · 2018-01-16
> **Re: Request for clarification**
>
> Thank you for the question, our wording is indeed not exact and will be amended in the final version of the paper. By "soft attention pooling" we refer to the graph-level output model in Equation 7 of [Li et al., 2015b], where the networks "i" and "j" are each a single fully connected layer with 128 output channels and tanh functions are replaced with the identity (as suggested in [Li et al., 2015b]).
> Our encoder implementation was based on https://github.com/mys007/ecc, where we adapted GraphPoolModule.py to use sum-pooling instead of max/mean-pooling. The gating itself can be implemented in a few lines:
>
> class SelfGate(nn.Module):
>     def __init__(self, lin1, lin2):
>         super(SelfGate, self).__init__()
>         self.lin1 = nn.Linear(64 + 4, 128)
>         self.lin2 = nn.Linear(64 + 4, 128)
>     def forward(self, input, input0):
>         inp = torch.cat([input, input0], dim=1)
>         return nnf.sigmoid(self.lin1(inp)) * self.lin2(inp)

---

### Public Comment · (anonymous) · 2018-01-16
**Question on the graph matching layer**

Interesting paper. How important is the graph matching layer to the whole network? There are recent graph matching methods that have been shown to outperform MPM (such as this one http://openaccess.thecvf.com/content_cvpr_2017/papers/Le-Huu_Alternating_Direction_Graph_CVPR_2017_paper.pdf). It is worth investigating whether replacing MPM by a better matching method will yield better results. It would be nice to include some discussion on this.

---

> ### Author Response · Authors · 2018-01-17
> **Re: Question on the graph matching layer**
>
> That's a good question. In Table 1 we show the performance of using no graph matching at all (NoGM), which works but produces many valid samples of low variety, so that we can claim that using some form of graph matching is certainly helpful. We have chosen MPM in particular because of convenience, as the algorithm is fast, and easy to understand and implement (also on GPU). There are indeed more recent graph matching algorithms (especially of higher order) but their implementation is either not public or is provided in Matlab (the case of your suggestion), which made them difficult and slow to integrate in our PyTorch codebase and thus, we have not run experiments with them. Nevertheless, we have found that 1) modifying the similarity function definition has played a more important role than changing parameters of MPM (such as number of iterations or trying out a sum-pooling variant), and 2) MPM scales relatively well with the size of the graphs (Table 2). While we cannot truly answer your question, our intuition is that a better graph matching algorithm would likely improve the results a bit but MPM itself does not seem to be the main performance bottleneck.

---

### Decision · Program_Chairs · 2018-01-29
**ICLR 2018 Conference Acceptance Decision**

**Decision:**

Reject

**Comment:**

The authors present GraphVAE, a method for fitting a generative deep model, a variational autoencoder, to small graphs.  Fitting deep learning models to graphs remains challenging (although there is relevant literature as brought up by the reviewers and anonymous comments) and this paper is a strong start.

In weighing the various reviews, AnonReviewer3 is weighed more highly than AnonReviewer1 and AnonReviewer2 since that review is far more thorough and the reviewer is more expert on this subject.  Unfortunately, the review from AnonReviewer1 is extremely short and of very low confidence.  As such, this paper sits just below the borderline for acceptance.  In general, the main criticisms of the paper are that some claims are too strong (e.g. non-differentiability of discrete structures), treatment of related work (missing references, etc.) and weak experiments and baselines.  The consensus among the reviews (even AnonReviewer2) is that the paper is preliminary.  The paper is close, however, and addressing these concerns will make the paper much stronger.

Pros:
- Proposes a method to build a generative deep model of graphs
- Addresses a timely and interesting topic in deep learning
- Exposition is clear

Cons:
- Treatment of related literature should be improved
- Experiments and baselines are somewhat weak
- "Preliminary"
- Only works on rather small graphs (i.e. O(k^4) for graphs with k nodes)